



# Modulation of the ENSO teleconnection to the North Atlantic by the tropical North Atlantic and Caribbean

Jake W. Casselman[1], Bernat Jiménez-Esteve[1], and Daniela I.V. Domeisen[1,2]

[1]ETH Zurich, Zurich, Switzerland
[2]University of Lausanne, Lausanne, Switzerland

**Correspondence:** Jake William Casselman (cjake@ethz.ch)

**Abstract.** El Niño Southern Oscillation (ENSO) can bring about inter-basin interactions, whereby Pacific sea surface temperature anomalies (SSTAs) may influence the North Atlantic European (NAE) region. However, ENSO's overall influence on the NAE remains unclear. One potential reason for this uncertainty may arise due to the region being dominated by several different mechanisms. Here we focus on one potential region, namely the Tropical North Atlantic (TNA), and determine how

the SSTAs modulate the ENSO teleconnection towards the NAE region. As numerous pathways from the TNA may exist for this modulation, we further center our analysis onto the Caribbean region and Walker cells. We force an idealized atmospheric circulation model with three different seasonally varying sea surface temperature patterns that represent an ENSO event with or without the influence of the Atlantic. Our results demonstrate that modulation of the NAE region by the TNA SSTA and Caribbean region occurs in the boreal spring and summer following an ENSO event. In boreal spring, this modulation is pri-

marily through a propagating Rossy wave train, while in the summer, the TNA's influence is nonlinear and tends to strengthen the ENSO influence over the NAE sector. Overall, this study offers a deeper understanding of the inter-basin interactions of the Walker cell following an ENSO event and the central role of tropical Atlantic SSTAs in modulating the teleconnection to the NAE region.

## 1   Introduction

El Niño-Southern Oscillation (ENSO) dominates interannual variability within the tropics and has global impacts through teleconnections. In addition to impacting the tropics, teleconnections can also influence extratropical regions, such as the North Atlantic European (NAE) region (Fraedrich and Müller, 1992; Fraedrich, 1994; Brönnimann, 2007). While it is not yet fully understood how ENSO impacts the NAE region, there is great potential for ENSO to create a relevant impact, which in turn may benefit long-range predictability over Europe (Domeisen et al., 2015). The complexity present in the ENSO-NAE

teleconnection results in part from several ENSO teleconnections influencing the region simultaneously. These pathways can be separated into several different parts, including signals through different geographical regions as well as different atmospheric levels (Rodríguez-Fonseca et al., 2016). For example, teleconnections can propagate through the extratropical troposphere (Bulić and Kucharski, 2012; Jiménez-Esteve and Domeisen, 2018; Mezzina et al., 2020), the extratropical stratosphere (Butler et al., 2014; Ayarzagüena et al., 2018; Domeisen et al., 2019), or the Caribbean region (Wulff et al., 2017; Hardiman et al.,





2019; Rieke et al., 2021). Furthermore, different regions modulate the regional response to ENSO teleconnections, such as the tropical North Atlantic (Sung et al., 2013; Casselman et al., 2021), and the Indian Ocean (Zhong et al., 2005; Fletcher and Cassou, 2015; Joshi et al., 2021). The teleconnections are, furthermore, sensitive to aspects such as the longitudinal position of Pacific sea surface temperatures (SSTs) (Zhang et al., 2019), non-stationarity (Rieke et al., 2021), and trends over time (Garfinkel et al., 2019).

Among the many influences on ENSO teleconnections towards the NAE region, the Tropical North Atlantic (TNA) influence remains unclear both in terms of the underlying mechanism, as well as the timing of the influence. The TNA region is salient for its strong connection to ENSO, whereby TNA SST anomalies (SSTA) are positively correlated with the Niño3.4 index and peak in boreal spring (March-May, MAM) following an ENSO event (Enfield and Mayer, 1997; Lee et al., 2008). This robust teleconnection travels through both a tropical and an extratropical pathway, influencing the TNA SSTs through either

moist convection/stability processes or changes in the trade winds (Jiang and Li, 2019; Casselman et al., 2021). There is also a connection between the TNA and NAE region. This connection includes the propagation of Rossby waves that can contribute to a circumglobal wave train (Toniazzo and Scaife, 2006; García-Serrano et al., 2011; Saeed et al., 2014; Scaife et al., 2017; Matsumura and Kosaka, 2019), as well as changes in the Hadley cell and Inter-Tropical Convergence Zone (ITCZ) (Okumura et al., 2001; Michel and Rivière, 2011). The resulting wave train can project onto different summertime circulation regimes

over the NAE region, such as the Atlantic low (Cassou et al., 2005).

The neighbouring Caribbean region also plays a role in the excitation and propagation of Rossby waves to the extratropics (Wulff et al., 2017; Hardiman et al., 2019; Neddermann et al., 2019; Rieke et al., 2021). In particular, Rossby wave trains originating in the Caribbean region can influence the Summer East Atlantic (SEA) pattern, impacting European summer climate (Wulff et al., 2017). A proposed forcing for this wave train includes opposing diabatic heating anomalies over the tropical

Pacific and Caribbean, referred to as the Pacific-Caribbean Dipole (PCD). The PCD is related to SSTAs between June and August (JJA), occurring during the developing phase of an ENSO event. The relationship between spring TNA/Caribbean SSTAs and the resulting wave train can also be utilized to improve seasonal prediction skill over Europe (Neddermann et al., 2019). However, the phasing of the resulting circumglobal wave train depends on a range of different aspects, therefore relying solely on TNA SSTAs is insufficient for anticipating the correct phasing (Neddermann et al., 2019).

In addition to the TNA SSTAs influencing the regional climate over Europe, the wave train launched from the tropics can continue as far downstream as Southwestern China, including an influence on rainfall and the East Asian summer monsoon (Li et al., 2018; Choi and Ahn, 2019). Often referred to as the East Atlantic/ West Russia (EA/WR) teleconnection, this teleconnection shows potential for skillful prediction in all four seasons (Lledó et al., 2020). Furthermore, since the EA/WR teleconnection may result from a westward extension of the downward branch of the Pacific Walker cell, and a Gill response

over the eastern equatorial Pacific, the EA/WR teleconnection may relate to the PCD index (Lim, 2015; Choi and Ahn, 2019).

It is clear that the Caribbean region and the associated PCD mechanism could create meaningful impacts on interannual timescales not just in the NAE region, but also eastwards of the NAE region. However, it remains unclear why the PCD and Caribbean Rossby Wave Source (RWS) are more related to the developing phase of an ENSO event. The precipitation dipole corresponds to an SST gradient, whereby the RWS intensifies when the Pacific and Atlantic basins have opposing SSTAs (Wulff





et al., 2017). As the PCD index locations also correspond well with the regional edges of the Pacific and Atlantic Walker cells, the precipitation dipole is also likely related to the climatological upwelling and downwelling branches of the two adjacent Walker cells, which can be influenced by underlying SSTs. Furthermore, since there is a co-variance between the Pacific and Atlantic SSTAs (Casselman et al., 2021), i.e. the basins often have anomalous SSTAs of the same sign, the role of the TNA for influencing the PCD following an ENSO event may be important. Specifically, it could be hypothesized that the positive

correlation between ENSO and the TNA SSTA may explain why the PCD is reduced/not active during the decaying phase of an ENSO event, as an intensifying TNA SSTA in boreal spring reduces the overall SSTA gradient between the basins.

The positive correlation between the Pacific and Atlantic SSTAs also results in key atmospheric differences related to different Gill Matsuno-type responses (hereafter shortened to Gill type response). In response to anomalously positive SSTAs and diabatic heating over the equatorial Pacific, tropical wave theory predicts that the atmosphere will be perturbed by creating a

pair of symmetric upper-level anticyclones (Gill, 1980; Lee et al., 2009). Downstream, the Walker cell is also perturbed via Kelvin wave propagation from the initial Gill type response, resulting in a Secondary Gill response. This response is opposite to the initial perturbation, creating descending motion and a pair of upper-level cyclones over South America, and is a key mechanism influencing the TNA SSTA (García-Serrano et al., 2017; Casselman et al., 2021).

Likewise, as the tropical Atlantic also tends to have a positive SSTA preceding an ENSO event, an atmospheric Gill-type

response is created, including a pair of upper-level anticyclones. However, given the nature of the tropical Atlantic SSTA (i.e., SSTAs predominantly occur north of the equator), the Northern Hemisphere exhibits an enhanced response, leading to an asymmetric Gill type response (Gill, 1980). Thus there is potentially a counteracting atmospheric response between the Secondary Gill response from an El Niño, and the asymmetric response created from the TNA SSTA. The main driver for perturbing the Atlantic Walker cell and thereby generating the asymmetric Gill response is also likely the Atlantic warm

pool, located in the Gulf of Mexico, Caribbean Sea, and western TNA, is a dominant source of diabatic heating (Wang et al., 2006, 2010; Rojo Hernández and Mesa, 2020). Climatologically, the influence of the Atlantic warm pool onto the Walker cell peaks during the warm pool's maximum extent, which occurs in boreal summer and fall.

However, when determining how ENSO teleconnections interact with different regions, a major issue for isolating influences onto ENSO teleconnection is the limited number of observed events in reanalysis datasets. As a result, several studies have

turned to model experiments to generate further events. Nevertheless, coupled models also have their own set of difficulties in representing ENSO variability correctly (i.e., SST diversity and strength), which is essential in recreating ENSO teleconnections properly (Frauen et al., 2014; Bayr et al., 2019). One solution to this problem is to prescribe realistic SSTs in an Atmospheric General Circulation Model (AGCM). By using an AGCM to isolate the influence from the TNA, this study aims to determine if the TNA modulates the influence on the ENSO teleconnection towards the NAE region. We consider the inter-

actions between the tropical basins, as well as interactions between the tropics and North Atlantic. We consider the differences between forcing the two basins simultaneously or separately to determine the linearity of the response and the areas where the Atlantic response acts constructively or destructively on the ENSO teleconnection towards the North Atlantic.



## 2 Methodology

### 2.1 Data

This study utilizes monthly mean SST from the Extended Reconstructed Sea Surface Temperature (ERSST) version 4 (Huang et al., 2015), as well as monthly mean fields (zonal/meridional winds, geopotential height, and precipitation) from the Japanese 55-year Reanalysis (JRA-55) (Kobayashi et al., 2015). We analyze the period from January 1958 until December 2019. We use a 30-year filter to remove low-frequency variability longer than 30 years for all fields, created using a fast Fourier transform (FFT). Statistical significance is calculated using a Monte Carlo test that is repeated 10,000 times to determine the significance

of the atmospheric response (Buckland and Noreen, 1990). This test selects a number of samples based on a given subsampling criterion, and compares the distribution to either the population from which the samples were drawn (as in reanalysis), or for the comparison of different model runs.

### 2.2 Model description

In addition to reanalysis data, we utilize Isca (Vallis et al., 2018), a simplified atmospheric general circulation model (AGCM),

to conduct a series of sensitivity experiments. This model uses the Geophysical Fluid Dynamics Labratory (GFDL) dynamical core, and uses the same model configuration as Jiménez-Esteve and Domeisen (2020). The atmosphere does not have explicit liquid water content but instead the moisture and radiative processes are considered using evaporation from the surface and a fast condensation scheme. Land-sea contrasts are created by changing the mixed layer depth, evaporative resistance, and albedo, while realistic topography is achieved by using the continental outline from ECMWF (Thomson and Vallis, 2018a, b).

Finally, the model resolution includes a Gaussian grid with 50 vertical levels (up to 0.02 hPa) and T42 horizontal resolution.

We perform three different sensitivity model experiments, along with a control run. The climatological SST forcing is created using the ERSSTv4 climatology from 1958-2019. In addition to the climatological SSTs, the sensitivity experiments include a Pacific-only (ENSO-like, 'P') forcing, Atlantic-only (TNA SST following an ENSO event, 'A') forcing, and a combined Pacific-Atlantic forcing, where SSTAs in both basins are forced ('AP'). Over the forced regions, the SST forcing is derived by

regressing the Niño3.4 index onto the SSTA field, which is then added to the background climatology. We choose to include both positive and negative SSTAs as previous studies have shown that the zonal SSTA gradient is important for influencing the Walker circulation (Zhao et al., 2021). The SST forcing is computed by regressing the October to February (ONDJF) mean Niño3.4 index onto the monthly SST field, backwards to August and forward to the following October. August to October of both the backward and forward regressions are averaged to create a smooth annual cycle. Similar to Jiménez-Esteve and

Domeisen (2020), regression values are multiplied by 4, resulting in an SSTA peak of approximately 4°C in the Pacific, and a peak of approximately 1°C in the Atlantic (for DJF and MAM, respectively, see figure 1 for forcing pattern and annual cycle). We chose to multiply the forcing by 4 to ensure there was a response, but it should be acknowledged that this forcing is stronger than in observations. Finally, for this study, we aim to determine the importance of the TNA following an El Niño only, and do not consider La Niña events.



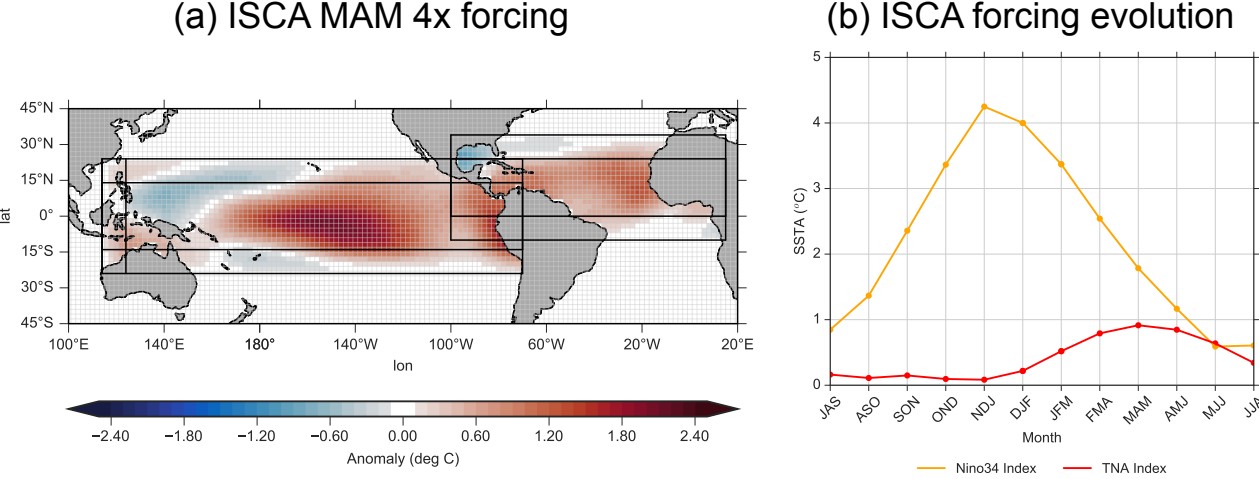

**Figure 1.** Forcing areas and seasonal evolution used to force ISCA. (a) depicts March-May (MAM) SST forcing for the AP runs (combined Pacific and Atlantic forcing). Black outlines indicate areas where regression was used, with outside rectangles indicating transition areas of linear decrease (see S4 for percentages). (b) shows the seasonal evolution of Niño3.4 (orange) and TNA (red) indices using a 3-month running mean.

The forcing region for the Pacific spans from the coastline of the Americas to 124°E, with a linear decrease to climatology from 124°E to 114°E. Similarly, in the meridional direction the forcing linearly decreases to zero between 14°N-24°N and 14°S-24°S. The Atlantic forcing spans the area from the African coast to the Americas. Since the dominant SSTA following an ENSO event is located in the northern part of the tropical Atlantic (peaking around 15°N), we use a linear decrease to zero from 0 to 10°S, and from 24°N to 34°N (Czaja et al., 2002).

Climatological runs are forced with monthly SSTs averaged between 1958-2019. There are 50 years with a recurrent SST seasonal cycle, of which the first 20 years are removed as spin-up. The forced SST experiments (P, A, AP) continue from the twentieth year of climatology for another 60 years. Removing the first year for spin up yields 59 years in total for each forced SST simulation. In order to compute the streamfunction and perform a Helmholtz-decomposition, missing values of the model experiments are filled solving Poisson's equation with a tolerance of $10^{-4}$, relaxation constant of 0.6, and an iteration

maximum of 89.

To remove the potential indirect influence from stratospheric variability, we relax (nudge) the zonal mean of the zonal winds above the tropopause to the daily climatology of the control simulation as in Jiménez-Esteve and Domeisen (2020). The relaxation term is computed at each model timestep and applied to the zonal mean zonal wind temporal tendency. Nudging is applied for pressure levels above 0.2 times the tropopause level and includes a transition layer between 0.5 and 0.2 times the

tropopause pressure level. See Jiménez-Esteve and Domeisen (2020) for the mathematical description.





## 2.3 Statistical Methods

We use multiple linear regression (MLR) analysis to isolate the linear influence from the Atlantic and Pacific, respectively, and plot the pointwise MLR coefficients following the method in Izumo et al. (2010) and Casselman et al. (2021). We also utilize partial correlation between three variables, following Wang et al. (2006). The partial correlation between variables X and Y, when removing the influence of variable Z, is calculated as follows:

$$R_{XY/Z} = \frac{R_{XY} - R_{YZ}R_{XZ}}{\sqrt{1 - R_{YZ}^2}\sqrt{1 - R_{XZ}^2}}, \tag{1}$$

where $R_{XY}$ represents the linear correlation coefficient between $X$ and $Y$.

## 2.4 Index definitions and Diagnostics

To evaluate Pacific SSTA, we use detrended 3-month running averages of SSTs averaged over the equatorial Pacific region bounded by 5$^o$N-5$^o$S, 170$^o$W-120$^o$W. The TNA index is defined as the area-averaged SST over the region of 5$^o$N-25$^o$N, 55$^o$W-15$^o$W, as in Taschetto et al. (2016). Finally, we utilize the Pacific-Caribbean Dipole (PCD) index as defined by Wulff et al. (2017). This index is determined by the normalized area-averaged precipitation difference between the Pacific and Caribbean (Pacific box [10$^o$N-20$^o$N, 180$^o$-110$^o$W] minus Caribbean box [10$^o$N-25$^o$N, 85$^o$W-65$^o$W]).

To analyze the upper-level circulation, we compute the streamfunction equation in Cartesian coordinates by using spherical harmonics (Dawson, 2016). This includes $u = \delta\psi/\delta y$ and $v = -\delta\psi/\delta x$, where $\psi$ represents the streamfunction, and $u(v)$ are the zonal (meridional) winds. The zonal mass streamfunction over the equatorial region is calculated analogously to the meridional mass streamfunction, except with a longitudinal and not latitudinal dependence, and we use the mean of winds between 5°N to 5°S. Furthermore, we utilize the Helmholtz-decomposed zonal divergence ($u_\chi$) for the streamfunction equation, under the assumption that the circulation is primarily thermally driven between 5°N to 5°S (Yu et al., 2012). The zonal stream function ($\Psi_z$) is defined as:

$$\Psi_z = \frac{2\pi a}{g} \int\limits_0^p [u_\chi]_{5N}^{5S} dp, \tag{2}$$

where $a$ is the Earth's radius, $g$ is gravitational acceleration, $p$ is pressure, and the brackets indicate the latitude for integration. An index measuring the Walker cell gradient between the Pacific and Atlantic, the *Walker index*, is defined as the upper level (200-400 hPa) streamfunction difference between the Atlantic (5°N-5°S, 240°-270°W) and Pacific (5°N-5°S, 310°-340°W) (Atlantic minus Pacific).

The Rossby wave source (RWS) (Sardeshmukh and Hoskins, 1988) is defined as:

$$RWS = -(\zeta\nabla v_x + v_x\nabla\zeta), \tag{3}$$

where $\zeta$ is the absolute vorticity, and $v_x$ is the divergent meridional wind. The first term represents vortex stretching, while the second represents vorticity advection. Furthermore, a Gaussian smoothing with a Gaussian kernel standard deviation of 0.7



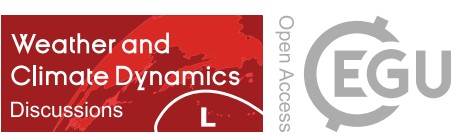

is applied to be able to interpret the large-scale responses. A localized *Caribbean RWS index* is defined as the area average of the 200 hPa RWS field bounded by 5-15°N, 80-60°W. Finally, we define the East Atlantic pattern as an area average of the 500 hPa geopotential height between 40-60°N, 310-340°E (see Supplementary figure S4). We chose this definition over an EOF as the dominant modes of variability change between the different sensitivity model experiments.

## 3 Results

### 3.1 Inter-Basin relationship between equatorial Atlantic and Pacific in reanalysis

We begin by comparing the different impacts from both the Pacific and Atlantic SSTAs onto the upper level circulation over equatorial South America and the Atlantic by applying an MLR analysis to the JRA-55 reanalysis dataset. Specifically, we aim to identify the respective influences each equatorial region may have on the Walker circulation. Understanding this interplay is salient as the combined influence over South America and the Caribbean region may be crucial for creating an RWS that can propagate a Rossby wave train that influences the extratropics. The MLR analysis allows for an approximate separation of the influences from the respective equatorial basins, which is necessary since ENSO and the TNA SSTA co-vary to a high degree. Finally, we focus on boreal spring and boreal summer for all fields as we are interested in the impact the TNA SSTA peak has following an ENSO event (MAM, Casselman et al. (2021)), as well as the time period when the Atlantic warm pool has the largest influence onto the Walker circulation (JJA). Both MAM and JJA are also related to wave trains towards the NAE region (Lim, 2015; Jung et al., 2017; Wulff et al., 2017; Li et al., 2018; Choi and Ahn, 2019; Neddermann, 2019; Lledó et al., 2020).

To better understand how each equatorial basin influences the vertical winds (associated with differing Gill type responses) over South America and the Caribbean, we show the boreal spring and summer Walker cells using an MLR with the zonal mass streamfunction and winds between 5°N and 5°S (Figure 2), in reference to the DJF Niño3.4 (preceding year) and MAM TNA (same year) indices. The streamfunction climatology (contours) over South America for both boreal spring and summer corresponds with the edges of the Atlantic and Pacific Walker cells, and ascending motion. Thus, by mass conservation, changes to vertical motion could easily modify the background upper level divergence, resulting in an RWS.

Anomalous Walker cell streamfunction and wind vectors for boreal spring and summer indicate that the influence from the Atlantic and Pacific SSTAs counteract one another during both boreal spring and summer. In boreal spring (Figure 2a-b) this counteraction includes strong upper level divergence over the eastern Pacific (associated with a weakening El Niño event), strong westerlies over South America, and descending motion around the western Atlantic. Conversely, Atlantic SSTAs are associated with divergence over the western Atlantic, and strong upper level easterlies over South America, consistent with Wang (2002). Notably, over South America and the western Atlantic there is counteracting descending and ascending anomalies associated with the Pacific and Atlantic SSTAs, respectively.

During boreal summer, anomalies related to both the TNA and ENSO largely continue from boreal spring, except that the upper level divergence due to Atlantic SSTAs has reduced, and vertical motion over South America has changed, whereby there is now descending motion around around the western Atlantic. As a result, shifting from boreal spring to boreal summer



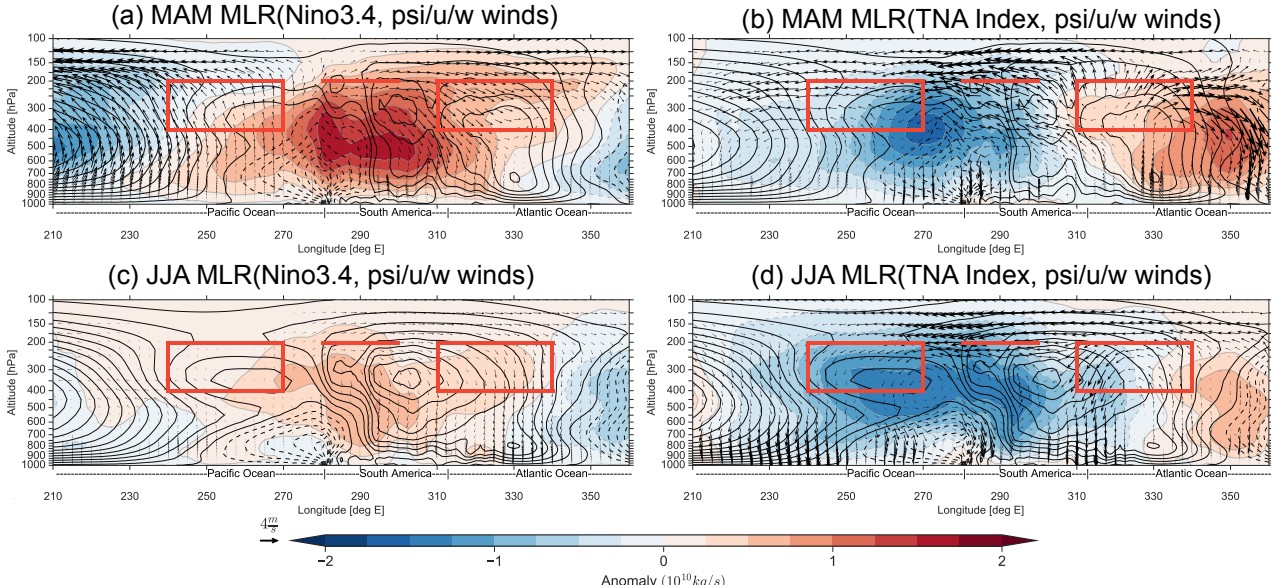

**Figure 2.** Pointwise multiple linear regression coefficients for the MAM (top) and JJA (bottom) zonal mass streamfunction (shading) and u and w winds components (vectors) on the MAM TNA SSTA and DJF Niño3.4 standardized indices. Contours represent the zonal mass streamfunction climatology (in intervals of 1.5 x $10^{10}$ kg/s), and winds are scaled by dividing the vertical velocity by 6.5 x$10^{-3}$. Red rectangles over the Pacific and Atlantic represent the Walker index, while the horizontal line at $290^o E$ represents the Caribbean RWS index. Data from JRA-55.

shows a shift from destructive vertical motion to constructive vertical motion over the western Atlantic and parts of eastern South America, while over central and western South America the result continues to show destructive interactions. As boreal summer streamfunction anomalies are much higher from the Atlantic forcing when compared to the Pacific, the Atlantic likely dominates the Walker cell perturbation. In contrast, in boreal spring, both basins perturb the Walker cell to a similar degree.

### 3.2 Lead-lag relationships of major tropic connections in Reanalysis

In the previous sections we found that there are counteracting upper level winds over South America as a result of the different forcings from the Atlantic and Pacific basins. Furthermore, vertical motion over South America and the Caribbean depends on the respective strength of the Atlantic and Pacific Walker cells, as this region sits at the boundary between both. This dependence on both cells may help explain why Wulff et al. (2017) could characterize the RWS over this region with a dipole index. As they also found that the precipitation field can be replaced by SSTs, this further indicates that both ocean basins play an important role in driving the PCD.

As we are interested in the respective influences from each equatorial basin onto the Walker cell, and the resulting upper level RWS over the Caribbean, we create a large-scale streamfunction gradient index between the Pacific and Atlantic, centered over



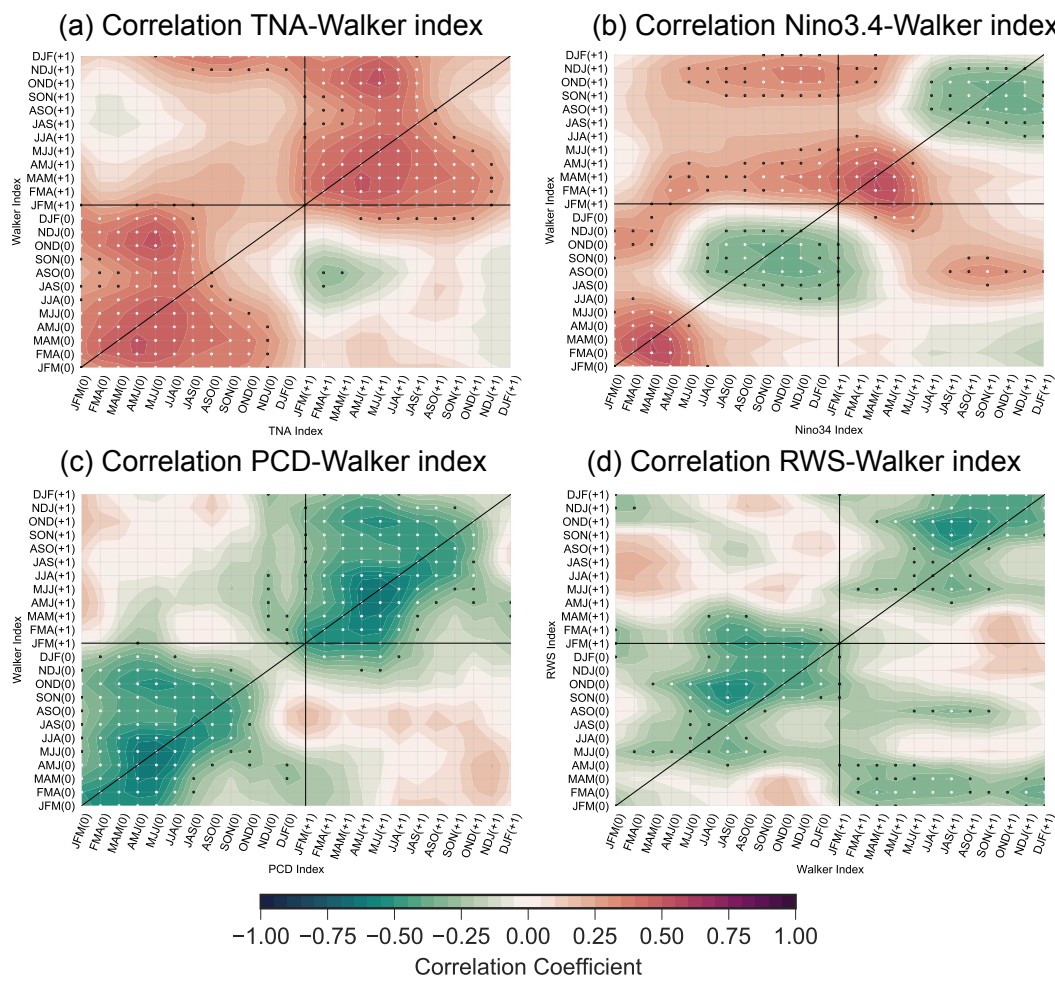

**Figure 3.** Lead-lag correlation between (a) the Walker index and TNA, (b) the Walker and Niño3.4 indices, (c) the Walker and PCD indices, and (d) the Walker and Caribbean RWS indices. '(0)' and '(+1)' refer to the starting year or the following year for each index, the diagonal line corresponds to no lag, black dots represent the 95% confidence interval, and white dots represent the 99% confidence interval. Results are using data from JRA-55 only.

the Caribbean. We complement this index with a more localized RWS index over the Caribbean. By using a lead-lag analysis, we further determine the dominant seasons for relating each indicate to the tropical basins. Figure 3 shows the 24 month lead/lag correlation of the Walker index with the TNA, Niño3.4, PCD, and the Caribbean RWS indices. The correlation between the TNA and the Walker index (Figure 3a) is positive, peaking in boreal spring at a significance level >99%. Furthermore, the

220 April-June (AMJ) to JJA TNA SSTA continues to influence the Walker index at the 99% level into the following boreal winter (DJF, 8 month lead). Similarly, the relationship between the Walker index and Pacific SSTA (Figure 3b) is positive and peaks in the following boreal spring, but becomes >99% significant with a 6 season lead for the Niño3.4 index (SON of previous year).





The relationship between the Walker index and peak of ENSO (boreal winter Niño 3.4) drops off rapidly after boreal spring, while the relationship with the peak of TNA SSTA continues until the following boreal winter. This indicates that the TNA SSTA dominates the Walker cell influence in boreal summer, when compared to ENSO, and is consistent with the previous MLR analysis. This relationship with the DJF Niño3.4 and Walker index also remains when using the boreal spring Niño3.4 (same season as TNA peak, MAM), as well as when considering a 0 lag correlation.

The boreal summer Walker index also has a negative correlation with the preceding ENSO event (Figure 3b), similar in timing (but opposite in sign, as the PCD and walker index dipoles are reversed) to the relationship found by Wulff et al. (2017) for the PCD-ENSO relationship. This hints at the similarities between the Walker and PCD indices. Thus, we further correlate the Walker and PCD indices (Figure 3c), showing that their relationship is significant above the 99% level for all seasons, with the exception of early boreal winter. The peak correlation occurs in boreal spring with a lead/lag of 0. As the relationship is continuous over most seasons and the indices measure similar aspects, the Walker cell and PCD index may be either highly coupled, or represent the same phenomenon. Finally, we relate the Walker index to the more localized Caribbean RWS index, which yields a similar relationship as the PCD index, with a negative correlation that is >95% significant during all seasons except in late boreal winter.

## 3.3 Tropical Pacific and Atlantic Interactions in Sensitivity Analysis Using ISCA

Up to this point, we applied an MLR to reanalysis data to determine the relationship between the Pacific and Atlantic SSTs with the Walker circulation. This is useful for showing that both basins play important roles, and also that the importance may shift between boreal spring and summer. However, this method can be misleading and limits our ability to explain nonlinear relationships as it assumes that the effects from each basin onto the Walker cell are additive (Osborne and Waters, 2003), which is likely not true. Due to the limited data in reanalysis and the high degree of correlation between ENSO and the TNA, it is also difficult to perform sensitivity experiments by subsampling based on whether the signal in the TNA SSTA is present following an ENSO event. To overcome these issues and to further test our MLR findings, we isolate the influences from the equatorial Pacific and Atlantic using the atmospheric model *Isca* with a series of sensitivity experiments.

In order to determine the respective and combined influence of both basins, we use separate forcings for the Atlantic ($A$), Pacific ($P$), as well as both basins forced simultaneously, i.e. Atlantic+Pacific ($AP$). We also look at the linear addition ($A+P$) of the responses from each basin in comparison to the simultaneous forcing ($AP$), and calculate the difference between forcing each region simultaneously or separately ($AP-(A+P)$). Finally, we also compare forcing an El Niño event that includes the Atlantic SSTA in the subsequent boreal spring to forcing the Pacific only ($AP-P$), thus isolating the Atlantic influence that often follows an El Niño event. It should be noted that this change should not be interpreted as the Atlantic response only (look to $A$ for this), but instead as the difference between $AP$ and $P$, as differences in the Pacific response may occur alongside the Atlantic response.

We consider the Walker cell between 5°N and 5°S, as in reanalysis, to determine the salience of the positive Atlantic SSTA, and its influence on the teleconnections arising from an El Niño event. Isca reproduces a realistic climatology (lined contours in figures 2 and 4), which includes strong clockwise (positive) cells over the central Pacific and Atlantic, and a small





counter-clockwise (negative) cell over the eastern Pacific. Notably, however, the boundary directly above South America is more disordered when compared to JRA-55 (contours in figure 2). This difference to reanalysis over South America may result from a poor representation of the low level flow over the Andes mountain range (i.e. a low level Kelvin wave response) due to

the comparably coarse model resolution (Rojo Hernández and Mesa, 2020).

Sensitivity experiments for boreal spring show that the Atlantic forcing *A* (Figure 4 a) creates a response similar to the JRA-55 MLR analysis, including strong ascending motion over the Atlantic, strong upper level easterlies over South America, and a large-scale counter-clockwise streamfunction anomaly. Also similar to our MLR analysis, the response to Pacific forcing *P* (Figure 4 b) creates divergence over the eastern Pacific, descending motion over South America and the western Atlantic

(Secondary Gill response), but lacks the strong upper level westerlies seen in the MLR analysis (figure 2a). When forcing both basins simultaneously (*AP*, figure 4 c), the large-scale streamfunction response matches the pattern when forcing the Pacific only (*P*), which includes a strong clockwise anomaly over South America and counter-clockwise anomalies over the Pacific and Atlantic. Thus, this similarity indicates that the Pacific remains the dominant influence onto the Walker cell during boreal spring for our model experiments, and that the Secondary Gill response from an El Niño likely dominates over the asymmetric

Gill response from the positive Atlantic SSTA. However, it should be noted that the Atlantic SSTAs do create a noticeable increase in upper level westerlies over South America, and strengthen the counter-clockwise streamfunction response over the eastern Pacific.

Forcing the Pacific and Atlantic SSTs both simultaneously and separately (Figure 4 c-d) yields a very similar response, indicating a high linearity between the responses. Both *AP* and *A+P* show an intensification of the upper level westerlies

over South America when compared to *P*. Further comparing *AP* and *A+P* (Figure 4 e) shows that wind anomalies are nearly identical and that much of the streamfunction differences over South America are not statistically different. Finally, we compare responses from forcing an El Niño only, with forcing an El Niño and Atlantic SSTA (*AP-P*, figure 4 f) and find that the difference closely resembles the *A* response. As the streamfunction over South America is not statistically different between *AP-P* and *A*, this further shows that during boreal spring, the influence from the Atlantic SSTA linearly interacts with the

influence that an El Niño event has with the Walker cell.

As we are also interested in how the influence shifts from boreal spring to boreal summer, figure 5 shows the Walker cell response to our sensitivity experiments for JJA. For *A* (Figure 5 a), the response for boreal summer compared to boreal spring shows similar upper level westerlies that extend into the Pacific, showing descending motion over parts of South America, consistent with our MLR analysis. At the same time, the response to *P* (Figure 5 b) largely lacks significant descending motion

over South America, showing that the Secondary Gill response has likely subsided.

When forcing the regions simultaneously (*AP*, figure 5 c) and comparing to *P* (Figure 5 f), the Pacific streamfunction response is weakened, while the South American streamfunction response is strengthened, corresponding well with the constructive/destructive areas in *A* and *P* (figure 5 a-b). *A+P* (Figure 5 d) also shows that the linear addition of each basin's influence results in a much stronger Pacific streamfunction response when compared to *AP* (Figure 5 c), indicating that the

Atlantic forcing may weaken the Pacific contribution over the Pacific. The difference between the simultaneous and separate



**Figure 4.** MAM Walker Cell analysis for the region connecting the Pacific and the Atlantic, averaged between $5°$S and $5°$N. Shading represents the streamfunction ($\psi$), contours represent the streamfunction climatology (in intervals of $1.5 \times 10^{10}$kg/s), and vectors represent the zonal winds ($u$), and vertical velocity ($w$) response to the different forcings from the Atlantic (a), Pacific (b), and Atlantic+Pacific combined (c), as well as the linear addition of the Pacific and Atlantic contributions (a+b). The contours represent the model climatology. The Atlantic contribution (e) is in addition derived as the difference between (c, AP forcing) and (b, P forcing), while the difference between the linearly combined response (A+P) and the combined response (AP simultaneously) is represented in (f). Stippling in (a-c) represents anomalies that are not statistically significantly different from climatology at the 95% level using a two-tailed Monte Carlo test. Stippling in (e) represents differences that are not statistically significantly different between the AP forcing (c) run and the linear combination of the Atlantic and Pacific (A+P, figure d), at the 95% level using a two-tailed Monte Carlo test. Stippling in (f) represents differences that are not statistically significantly different from the A forcing run at the 95% level using a two-tailed Monte Carlo test. The rectangular boxes represent the areas for computing the Walker index, while horizontal line represents level and longitude for RWS index.

**Figure 5.** Same as figure 4, but for JJA.

forcing (*AP-(A+P)*, figure 5 e) also shows statistically significant differences in the streamfunction response over the Pacific, while the upper level wind response is much more linear.

To expand on the influence from adding the Atlantic SSTA, figure 5 f shows the difference between forcing an El Niño event with and without the Atlantic forcing (*AP-P*). Results show a spatially similar, but significantly stronger streamfunction

response when compared to both the *A* response, as well as the *AP-(A+P)* difference. This close resemblance in terms of spatial pattern potentially indicates that the nonlinearity arises from an intensified Atlantic influence, with respect to forcing the Atlantic independently. Overall, for the boreal summer, we find that the Atlantic contribution continues to dominate the upper level westerlies over South America and now the overall streamfunction magnitude is influenced by the Atlantic SSTA. Furthermore, the nonlinearity likely arises from an intensified Atlantic response when coupled with the Pacific SSTA.

In addition to considering the Walker cell for understanding the competing Gill type responses (symmetric vs. asymmetric), we utilize the 200 hPa streamfunction. We again consider the same sensitivity experiments, and differences between responses



**Figure 6.** MAM 200 hPa streamfunction (shading) and non-divergent winds (vectors) analysis for the response to the different forcings from the Atlantic (a), Pacific (b), and Atlantic+Pacific combined (c), and linear addition of the Pacific and Atlantic contributions (a+b). The Atlantic contribution (e) is derived as the difference between (c, AP forcing) and (b, P forcing), while the difference between the linearly forced response (A+P) and the combined response (AP simultaneously) is represented in (f). Stippling in (a-c) represents anomalies that are not statistically significantly different from climatology, stippling in (e) represents differences that are not statistically significantly different between the AP forcing (c) run and the linear combination of the Atlantic and Pacific (A+P, figure d), and stippling in (f) represents differences that are not statistically significantly different from the A forcing run, all at the 95% level using a two-tailed Monte Carlo test. Black box represents the RWS index over the Caribbean region.



**Figure 7.** Same as figure 6, but for JJA



to determine linearity. Beginning in boreal spring, the response to *A* (Figure 6 a) shows an anti-cyclonic maximum over the Caribbean, and a dipole centered about the equator with a dominant northern section (asymmetric Gill response, consistent with Gill (1980)). Similarly, the response to *P* (Figure 6 b) depicts a dipole over the deep tropical Atlantic consistent with García-Serrano et al. (2017), but the dipoles merge with the midlatitude response instead of forming distinct cyclones.

The response when forcing the Pacific and Atlantic simultaneously (*AP*, figure 6 c) shows a strong resemblance to the *P* response, but for *AP* the anti-cyclonic activity over the Caribbean (associated with the Atlantic SSTA) is intensified, and the Secondary Gill response from ENSO is more distinct from the midlatitude response. As little difference exists in comparison to the linear combination (*A+P*, figure 6 d), the symmetric and asymmetric Gill responses from each basin add linearly in boreal spring. Figure 6 e further shows that there is little difference in the tropics between the simultaneously and separately forced responses (*AP - (A+P)*), since no clear spatial patterns occur that are statistically significant. The high linearity in this relationship can also be seen in figure 6 f, as the change in the response created by adding of the Atlantic forcing to an El Niño event closely resembles the tropical response from *A*. However, this cannot be said for the extratropical response, which will be discussed in the forthcoming subsection.

Shifting to boreal summer, figure 7 shows that the *A* and *P* responses (Figure 7a-b) result in generally opposite influences in the northern hemisphere, and that when combined in *AP* (Figure 7c), the asymmetric Gill response from the Atlantic dominates. By comparing the simultaneous and separate responses (*AP* vs. *A+P*), we see a nonlinearity arising, whereby the asymmetric Gill response is stronger in *AP* than in *A+P* (also see figure 7e). One potential reason for this change can be seen in figure 7f, which shows that the difference in response by adding the Atlantic SSTA forcing to an El Niño event is a stronger and more confined asymmetric Gill response over the Caribbean/South America than *A*. Overall, this nonlinearity in the tropics is consistent with the Walker analysis (Figures 4 and 5) where the inter-basin relationship shifts from linear to nonlinear from boreal spring to boreal summer. It is also consistent with our lead-lag analysis showing that the dominant influence perturbing the Walker cell shifts from the Pacific to Atlantic SSTAs from boreal spring to summer. As *AP-P* represents the overall result of adding the Atlantic SSTA, this nonlinearity may arise through an intensified Atlantic response in boreal summer, a weakened Pacific response, or a combination of both. However, given the spatial pattern matches the forcing from *A*, it is likely an intensified Atlantic response.

### 3.4 Modulation of the tropical connection towards the North Atlantic-European Region by the Caribbean

Previously, the interaction between the tropical basins interested us as the combined influences may create a potential RWS that can influence the NAE region. As we now have a better idea of this inter-basin interaction, we next investigate how the Atlantic SSTA may influence the NAE region during an ENSO event via a Caribbean RWS. To do this, we quantify the aforementioned tropical interactions using the Caribbean RWS index and Walker index, while we use the East Atlantic (EA) pattern and North Atlantic Oscillation (NAO) to quantify the response in the NAE region. Consistent with previous sections, we again look at boreal spring and summer.

The Walker and Caribbean RWS indices have a strong relationship that varies considerably between sensitivity runs in boreal spring (Figure 8 a). For *P*, the correlation is -0.28, while for *A* it is -0.33. When forcing both basins together (*AP*), the correlation





increases by 57% to -0.44 as compared to only forcing the Pacific basin ($P$). The overall correlation is also relatively similar for boreal summer (Figure 8d), except that for $P$ and $AP$ the correlation is similar (-0.53 and -0.55, respectively), indicating a weaker impact of the Atlantic SSTA in boreal summer.

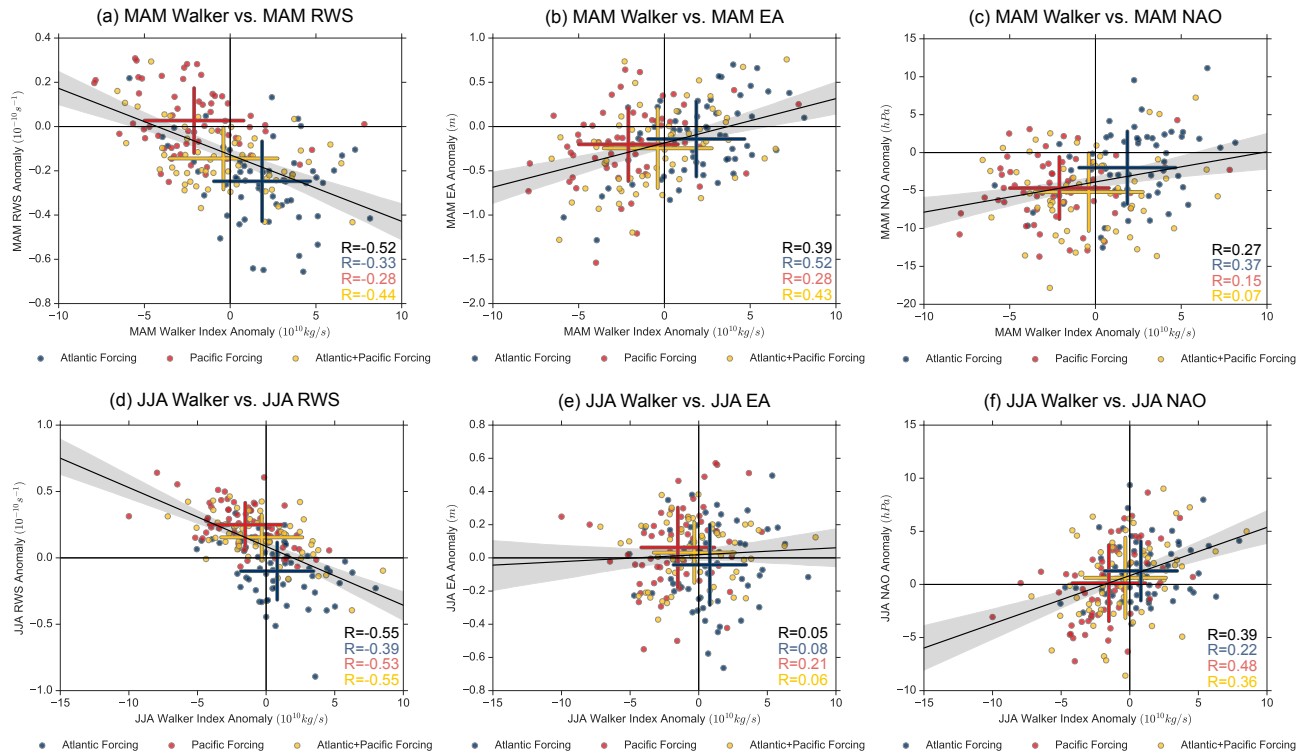

**Figure 8.** Scatter plot for model data (forcing area indicated in legend) showing the key relationships between the Walker index, RWS index, East Atlantic index, and NAO index. The correlation coefficient ($R$) is represented in the bottom right corner of each panel, with colors representing the forcing area and black as the $R$ when using all data points. Crosses represent the mean (center), and standard deviation in x and y directions. Shading represents the 95% confidence interval.

The relationship between the sensitivity experiments and Walker index is constant for both boreal spring and summer (Figure 8a,d x-axis), including a positive anomaly for $A$ and a negative anomaly for the $P$ responses. In contrast, the Walker response to $AP$ forcing is approximately 0 for both seasons. However, $AP$ exhibits a large Walker index variance, whereby the anomalies cover much of the range of $A$ and $P$ (see the yellow cross in Figure 8a,d). Thus, this indicates that the Atlantic SSTA can perturb El Niño's influence onto the Walker gradient in boreal spring and summer with an anomaly that is approximately equal and opposite of the Pacific's influence, resulting in a neutral gradient for $AP$.

Overall, for the Walker index-RWS relationship, the boreal spring Caribbean RWS index for $AP$ and $P$ become significantly different (>95%) as the RWS anomaly moves from relatively neutral for $P$ to negative for $AP$ (Figure 8a). Since the mean RWS for $P$ is neutral even with a strongly negative Walker gradient, this indicates that the Walker gradient is not the sole influence on





the Caribbean RWS. Still, the correlation increases markedly for *AP*, showing that the overall relationship between the Walker index and RWS is strong when forcing the regions more 'realistically' (i.e., including the Atlantic SSTA that often follows an

El Niño event). Furthermore, as the correlation between the Walker and Caribbean RWS indices increases notably into boreal summer for *P*, the mean RWS for *P* also becomes positive.

The relationship between the Walker and EA index also varies considerably between sensitivity experiments but generally shows a stronger relationship in boreal spring (Figure 8b,e). For boreal spring, the introduction of an Atlantic SSTA (*A* has an R =0.52) increases the correlation between the Walker index and EA index from 0.28 for *P* to 0.43 for *AP* (54% increase). This

large increase indicates the importance of including the Atlantic during an El Niño event (*AP*) during boreal spring. However, since the mean EA anomaly for all sensitivity runs is relatively similar, at around 0 to -0.5 m, this results in little change in the EA anomaly. Conversely, in boreal summer, the correlation is insignificant for both *P* and *AP*. Furthermore, forcing both basins together results in a decrease in the correlation from 0.21 to 0.06 for *P* and *AP*, respectively.

Correlation for boreal spring and summer Walker index and NAO (Figure 8c,f) shows that when the Atlantic SSTA is added

(i.e., going from *P* to *AP*), the correlation drops. This decrease indicates that the Atlantic SSTA modulates that connection by weakening the Walker cell and NAO relationship. As a result, the correlation is insignificant in boreal spring, while in boreal summer, we see a statistically significant correlation between the Walker index and NAO.

To further explain the importance of the Caribbean RWS for influencing the NAE region, we use a partial correlation to remove the influence of the proposed mediating pathway (the Caribbean RWS) from the Walker index-EA/NAO connection

for each sensitivity experiment (Figure S3). Bivariate correlation shows that the RWS connection to the EA/NAO also changes between *P* and *AP*, whereby the *AP* shows a notable increase in the NAO connection in boreal spring, resulting in both the RWS-EA/NAO connections being largest in boreal spring. When removing the RWS influence using a partial correlation in boreal spring for both *P* and *AP*, the Walker index-EA correlation decreases notably, while the Walker index-NAO correlation either increases slightly or has a negligible change (Figure S3, solid vs. dashed red and green lines). Overall, this indicates

that the Caribbean RWS may mediate a potential connection between the Pacific and Atlantic Walker cells and the EA during boreal spring and the NAO during boreal summer. However, as the partial correlation does not remove all correlation, other external factors may be at play.

Next, to better understand the NAE regional response, we determine the spatial anomalies in the extratropics by plotting the boreal spring MAM 200 hPa geopotential height response (Figure 9). The *A* response (Figure 9a) is associated with a

meridional wave train from the tropics to mid-latitudes that curves eastwards in the extratropics. This response is similar to the EA/Russia pattern found by Lim (2015), which was found to be related to Rossby wave propagation from the tropics. Other studies have also associated this pattern with Atlantic SST variability (Jung et al., 2017; Li et al., 2018; Choi and Ahn, 2019; Lledó et al., 2020).

Forcing the Pacific only (*P*, Figure 9b) yields a largely negative response over the midlatitudes (around 45°N, 30°W), which

is generally constructive to the *A* response. Comparing *AP* and *A+P* shows that the responses appear to add relatively linearly. However, when determining the difference between *AP* and *A+P* (Figure 9e), we see a statistically significant positive EA pattern. The EA pattern in figure 9e indicates that when forcing the two basins simultaneously, the EA pattern is less present



**Figure 9.** Same as figure 6, but for MAM 200 hPa geopotential height (shading) and 200 hPa irrotational winds (vectors).



**Figure 10.** Same as figure 9, but for JJA.





during *AP* than when linearly adding the two basin forcings (*A+P*). As figure 9e resembles the negative response from *A*, the nonlinearity seen in 9e is likely due to a weakened Atlantic response. The overall Atlantic response also alters the wave
propagation and creates a more zonal-like structure in the NAE region (Figure 9f). As a result, there is little overlap of the overall Atlantic contribution with both the EA and NAO modes. This may explain why we find that El Niño events perturb both the EA and NAO to similar amounts for both *P* and *AP* (see figure 8b-c).

For boreal summer (Figure 10) the 200 hPa geopotential height response for *A* (Figure 10a) lacks a wave train towards the NAE region. Instead, a strong meridional dipole is present between the Mediterranean and Northern Europe. For *P* forcing
(Figure 10b), much of the response over the NAE region is insignificant. The exception to this is a significant positive anomaly over the North Atlantic and a dipole over Europe matching the *A* response in polarity, but with the southern lobe being the only significant area. *AP* (Figure 10c) shows a response that resembles the *P* response much more than the *A* response in the extratropics, indicating that the Pacific response remains the dominant influence in boreal summer over the NAE region. This resemblance to *P* is in contrast to the response for the linear addition (*A+P*), where the dipole shows up more clearly due to
the linear combination of *A* and *P*.

The difference between *AP* and *A+P* (Figure 10e) shows that the NAE anomaly resembles *A*, but with the opposite sign. Thus the difference between *AP* and *A+P* likely occurs due to a weakened Atlantic response in the NAE region. This reduced strength is shown in figure 10f, where the addition of the Atlantic SSTA forcing results in a weaker response when compared to *A* alone. Furthermore, the *AP-P* difference appears to diminish north of the subtropical region, including a significantly
reduced negative anomaly over Northern Europe, when compared to *A*. Overall, independently, the Atlantic creates a robust dipole signal in the NAE region (as seen in *A*), but when added to an El Niño event, this influence is reduced considerably (Figure 10f). Namely, the influence cannot travel as far north, making the Pacific remain the dominant influence on the NAE region into the boreal summer. The Atlantic SSTA primarily modulates an El Niño's response by primarily reinforcing the dipole over Europe, as both lobes are now significant.

**4   Discussion and Conclusion**

Using MLR analysis and a series of sensitivity experiments, our findings suggest that the tropical North Atlantic modulates the ENSO teleconnection towards the NAE region. To explain the mechanism of this modulation, we first focus on breaking down the tropical interactions, analyzing the Walker cell, and the interactions between the asymmetric and symmetric Gill responses. As far as we are aware, studying the interrelation of Gill responses over the tropical Atlantic is a novel consideration, building
on the growing interest in inter-basin interactions and the newly discovered Secondary Gill response by García-Serrano et al. (2017). Furthermore, we hypothesize that the Walker cell interactions between the basins explains the underlying mechanism driving the PCD index (Wulff et al., 2017). The PCD is salient for its ability to propagate to the NAE region via a Rossby wave train, and a potential pathway for the Atlantic to influence the NAE region.

Indeed, by quantifying the Walker circulation using the zonal mass streamfunction over the Pacific and Atlantic regions, we
show that the PCD is highly coupled to the Walker cell index. Increasing our physical understanding of how the Atlantic SSTA



relates to the PCD index can be instrumental for studies that seek to improve seasonal predictions by using the PCD index, such as those by Neddermann et al. (2019). Additionally, by using a series of sensitivity experiments, we show that the Pacific dominates the tropical interaction in boreal spring. The Atlantic contribution is a modulation of the Walker cell, especially the upper level horizontal winds over South America. When the Secondary Gill response largely subsides by boreal summer, the

Atlantic influences the Walker cell streamfunction more, and also begins to become nonlinear. Thus, our results cast a new light on how hemispherically symmetric and asymmetric Gill responses interact linearly in boreal spring, but nonlinearly in boreal summer. The source of this nonlinearity may be an intensified Atlantic response in the presence of an El Niño event. This finding can be particularly valuable for studies looking to understand the downstream influence of the tropical Atlantic on aspects such as rainfall and monsoon patterns, as the strength of the Atlantic Walker cell may change in the presence of a

Pacific forcing (Kucharski et al., 2009; Wang et al., 2009).

When determining the inter-basin interaction on the Walker circulation, using an AGCM may have inherent limitations in properly resolving the Walker circulation. It could be argued that the horizontal model resolution remains a key limitation. One area where this may be limiting our study is heat balance and the atmosphere's response to SST gradients, which are important for driving the Walker circulation. Models with higher resolution are better able to balance the response through the transient

atmospheric circulation (Parfitt et al., 2016). Utilizing a moist model with fast condensation that lacks explicit liquid water may also limit the accurate representation of the Walker circulation in addition to model resolution. This limitation centers around properly modeling the precipitation and divergence above the tropical Pacific during an El Niño event, which influences Kelvin wave propagation. Downstream, the Kelvin wave can also be influenced by difficulties in representing South American rainfall, which is a major issue in several AGCMs, as well as many coupled models (Gudgel et al., 2001; Maher et al., 2018; King et al.,

2021). As a result of an altered Kelvin wave, this may limit the accuracy of the Secondary Gill response. Nonetheless, given that the location of both the Secondary Gill and asymmetric Gill responses match those from reanalysis (Casselman et al., 2021), we believe that our results are representative of the observations.

Although from our results, it is clear that the Pacific dominates the Walker circulation during boreal spring, the Atlantic modulates the Walker index considerably, whereby the correlation with the Caribbean RWS nearly doubles when adding the

tropical Atlantic forcing to the Pacific forcing (*AP* response). It is notable that although the Walker index and RWS have a strong relationship, the linear fit does not pass through the origin, whereby the *P* has a strongly negative Walker index but lacks an anomalous RWS. This incongruency is in line with studies that have struggled to reconcile the relationship between the Caribbean RWS and ENSO, indicating that ENSO may come before, after, or not be related at all to the RWS (and associated wave train) (Ding and Wang, 2005; Wulff et al., 2017; Neddermann et al., 2019). Given the relative importance of the Atlantic

SSTAs in improving the relationship between the Walker and RWS indices from *P* to *AP*, had previous studies changed the influence of the Atlantic, they may have observed very different results. Therefore, the modulation of the Atlantic SSTA onto the Walker cell is important and should be considered when determining the relationship between the Caribbean RWS and an ENSO event.

We also investigate the Caribbean RWS impact on key modes of variability in the NAE region, namely the EA and NAO

patterns. A key finding includes that the Walker index is highly related to the EA pattern in boreal spring and the NAO pattern in



boreal summer. Bringing this further by using a partial correlation to remove the hypothesized mediating pathway (Caribbean RWS), we show that the RWS may indeed be the mediating pathway. However, as the correlation between the Walker index and Caribbean RWS increases very little by adding the Atlantic SSTA in boreal summer (-0.53 to -0.55), this indicates that the Atlantic modulates this connection little. Thus, the dominant modulation of the extratropical response via the tropical Atlantic

SSTA and a perturbed Caribbean RWS occurs in boreal spring, primarily for the EA pattern only.

Examining this finding further, very different extratropical geopotential responses are found between boreal spring and summer. A wave train from the tropical Atlantic to the extratropics is only present during boreal spring. The fact that the wave train from the Caribbean to Europe is only present in boreal summer may be due to changing background conditions. Supplementary figure S2 shows the background 200 hPa absolute vorticity and zonal winds, which are useful for estimating

wave guide locations. Figure S2c-d indicates that the a separation of the subtropical and eddy driven jets changes between boreal spring and summer, whereby a clear separation is only present in boreal spring. As a result, the increased separation of the jets or intensification of the eddy driven jet may create more suitable conditions for the propagation of the wave train from the Caribbean.

Further analysis of the boreal spring wave train shows that it overlaps with the EA pattern, which may explain why the

correlation between the Walker and EA indices increases significantly when adding the Atlantic influence but diminishes it for the NAO index. However, as the wave train from the Atlantic changes considerably (i.e., weakens and shifts) in the presence of the Pacific SSTA, the Atlantic influence onto the EA changes between *A* and *AP*. Given that this occurs even as the Walker cell relationship is found to be linear in boreal spring, the source of this reduced extratropical response is likely not related to the tropical interactions.

In boreal summer, the extratropical response to the Atlantic forcing is reduced when forced together with the Pacific (in comparison to *A*), but the tropical Atlantic response is intensified when forced alongside the Pacific. Therefore, it remains unclear what causes the nonlinearity in the extratropical response during boreal summer. The nonlinearity may be due to weakened wave propagation in boreal summer or the outcome of weakened propagation occurring earlier in boreal spring, with the residual dipole in boreal summer being the overall result (see figure S5 for the evolution of the residual). However,

when considering atmosphere-ocean interactions in the extratropics, there are some key limitations presented when using a series of sensitivity experiments in an AGCM. The first limitation is that we rely on prescribed SSTs, removing the ability of the ocean to respond to atmospheric processes and also changing thermal damping effects (Barsugli and Battisti, 1998). This simplification is essential in isolating the influence from the tropical Atlantic during an El Niño event but may create issues in the extratropical regions. Here, the NAO is the dominant mode of variability in the NAE region that may act back onto the

ocean, in the form of the SST tripole pattern (Peng et al., 2003). However, Baker et al. (2019) found that the tropical Atlantic SSTs are sufficient for properly predicting the NAO response and that the lack of atmosphere-ocean coupling in the North Atlantic does not impact the predictability.

Our results provide a basis for understanding the importance of the TNA SSTA by answering several key questions, but they also bring light to several currently unanswered questions. In this study, by multiplying the forcing by a factor of 4, we assume

that the TNA SSTA scale linearly with the strength of ENSO. However, as Casselman et al. (2021) showed, the TNA SSTA



may begin to plateau in strength with extreme ENSO events. As a result, one would expect any influences from the TNA SSTA to be reduced during extreme events. This interaction may also increase in importance in the future if climate change influences the strength of ENSO events (Cai et al., 2021). As this study only considers El Niño events, future studies should determine if the negative TNA SSTAs that result from a La Niña interact similarly with the Walker gradient. This may be especially

important if the wave activity propagates into the NAE region differently between the positive and negative phases of ENSO (Feng et al., 2017).

As we intentionally excluded ENSO teleconnections that might be present in the stratosphere via a nudging factor, this has reduced the sensitivity of the NAO to ENSO. Jiménez-Esteve and Domeisen (2020) found that the interannual variability of the winter NAO variance decreased by 40% when including this nudging, which may result in misrepresenting the importance of

the TNA for influencing the NAE region. However, as the stratospheric pathway is likely weaker in MAM and JJA as compared to mid-winter or early spring, we do not expect that removing the stratospheric influence in our study should change the NAE response in a major way. Nonetheless, Future studies should determine the importance of the TNA SSTA for modulating the full influence from ENSO through all pathways.

*Code and data availability.*  To access the JRA-55 and ERSSTv4 datasets, they are available at the NCAR data archive (https://rda.ucar.edu/).

The Isca modeling framework was accessed from the GitHub repository (https://github.com/ ExeClim/Isca, last access: May 2020) (Vallis et al., 2018).

*Author contributions.*  JWC performed the model simulations, analysis, and writing of the manuscript. BJE contributed to the setup of the AGCM and to editing the manuscript. DIVD contributed to analysis and interpretation of the results as well as writing the manuscript.

*Competing interests.*  The authors declare that they have no conflict of interest

*Acknowledgements.*  We would like to thank Andrew Dawson for creating the gridfill and windspharm python packages. The work of J.C. is funded through ETH grant ETH-17 18-1. This project has received funding from the European Research Council (ERC) under the European Union's Horizon 2020 research and innovation programme (grant agreement No. 847456). Support from the Swiss National Science Foundation through projects PP00P2_170523 and PP00P2_198896 to D.D. is gratefully acknowledged.





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
