# Peer review of "Modulation of the El Niño Teleconnection to the North Atlantic by the Tropical North Atlantic during Boreal Spring and Summer"

_Weather and Climate Dynamics, 2021_

## Author Comment (AC1)

Dear Dr. William Roberts,

We would like to express our gratitude to the editor and reviewers for their time and work in evaluating our paper. The ideas aided in further improving the research. Additionally, we identify the significant modifications we made to the paper throughout the editing process. We have addressed the following points in particular:

- The introduction was reduced and made more succinct
- Our methods now include the Fisher z-transformation, improving the statistical robustness of our results
- We further expanded on the caveats of using a 4x forcing in our models
- Sections 3.1 and 3.2 were made more succinct and refined with further explanations for why the reader should care
- Sections 3.3 and 3.4 were improved by changing the significance test in subfigure (f) to be (1) more consistent with the all subfigures, and to (2) outline the significant differences better between P and AP experiments
- Section 3.4 was improved and edited to include the Fisher z-transformation confidence intervals and to indicate where the Atlantic SSTAs both significantly and insignificantly modulate ENSO's connection to the NAE region

Please find the detailed responses to the reviewer's comments and suggestions below.

Jake Casselman, on behalf of all authors

REVIEWER COMMENTS:

All changes in the text have been **boldened**, and explanations for each comment are stated below the comment in **blue** writing.

**Reviewer #1:**

SUMMARY:

The study is about inter-basin interaction of the Walker Cells and Gill responses forced by ENSO-related SST in the Pacific and Tropical North Atlantic during boreal spring and summer, as well as their teleconnection to the North Atlantic. Analyses from reanalyses and atmospheric model experiments are performed.

I think this is an interesting and important topic. However, currently, I find that the main problem is the presentation/writing lacks connections among the every smaller piece of results described. This is perhaps due to the reason that there is a lack of explanations on how the different pieces fit together. The paper in some parts reads like a mere listing of many features found, and it is easy to become lost for the readers. I describe this issue in a number of specific subsections below.

I also have some questions about clarifying what relationships are being modulated and what are the modulators.

**MAJOR COMMENTS:**

Sects. 3.1, 3.2: There are various features and properties identified and described. I find it somewhat difficult to digest because I am not quite clear about what are the reasons for highlighting those specific features. I suggest to consider the main aims of the study when selecting features that are essential to focus on, and tell us also exactly why we should look at them in relations to the central questions of the study and how they link to the next subsections.

Section 3.1 begins by answering the first part of our goals, namely the tropical interactions, by considering how the TNA and ENSO interact in reanalysis. We state this aim: "identify the respective influences each equatorial region may have on the Walker circulation" (line 178). These results from reanalysis later serve as a comparison when using our model in order to understand how the basin interactions are represented in the model. To improve section 3.1, we have removed the sentences between lines 190-192 as they do not directly address this goal, allowing us to combine paragraphs 2-3, making section 3.1 a succinct 3 paragraph description of reanalysis.

Subsection 3.2 aims to address the differences in timing between the influences of each tropical basin on the tropical circulation and to introduce and justify the Walker cell index and Caribbean RWS index by showing that there is a significant connection between both indices. To make the subsection clearer we have done several adjustments:

- We have moved "**As we are interested in the respective influences from each equatorial basin onto the Walker cell, and the resulting upper level RWS over the Caribbean, we create a large-scale streamfunction gradient index between the Pacific and Atlantic, centered over the Caribbean. We complement this index with a more localized RWS index over the Caribbean.**" into first paragraph to help move the justification closer to the beginning.

- We rearrange the justification sentence from "By using a lead-lag analysis, we further determine the dominant seasons for relating each indicate to the tropical basins." towards: "**We use a 24-month lead/lag correlation analysis to analyze the Walker cell relationships, and to further understand how connections with ENSO and the TNA change between MAM and JJA.**"

Sect. 3.3: This is again a rather long catalogue of different combinations, comparisons and observations of the model results. It is not described clearly why all the different results should be of interest in connection with the central themes. Yes, ok, there is a linear constructive/destructive effect, there is also some nonlinear effect in certain features, and there are some agreements and disagreements with the reanalysis data. But try to tell us why they are important in their own right and/or for the central themes.

Thank you for your feedback on this subsection.

Subsection 3.3 pertains to tropical inter-basin interactions, and follows the goal stated at the end of the introduction (First, to analyze the tropical modulation of the TNA on the ENSO teleconnections, then how this modulation influences the extratropical connection of ENSO onto the North Atlantic. We further place key attention on the linearity, and the timing, as well as contrasting reanalysis and our sensitivity model runs (outlined on lines 89 to 92)):

We describe the main rationale behind considering the results in the following:

- Why should we look at model data and not reanalysis (lines 238-245)
- Why we should consider A, P, AP, A+P, AP-(A+P), and AP-P (lines 246-253)
- Why we look at the Walker cell (lines 254-255)
- Why we also use the streamfunction (lines 300-301)

To make the subsection feel less "catalogue"-like, we improve the manuscript by including the following:

- Improving explanation for moving from an MLR to a model by changing "**However, this method can be misleading and limits our ability to explain nonlinear relationships as it assumes that the effects from each basin onto the Walker cell are additive…**" To: "**However, this method cannot fully capture the nonlinear contribution, as it assumes that the effects from each basin onto the Walker cell are additive**..."

- In order to further state the reasons for looking at A, P, AP, A+P, AP-(A+P), and AP-P, we edit the section to now read: "In order to **address the sensitivity of ENSO to the TNA SSTAs, we** determine the **separate** and combined influence of both basins **using** separate forcings for the Atlantic (A), Pacific (P), as well as both basins forced simultaneously, i.e., Atlantic+Pacific (AP). **To determine the linearity of ENSO's response to the addition of the Atlantic SSTAs,** we also look at the linear addition (A+P) of the responses from each basin in comparison to the simultaneous forcing (AP), and calculate the difference between forcing each region..."

- To further justify the streamfunction composites, we add the explanation that "**As the Secondary Gill and asymmetric Gill type responses have distinct**

**spatial patterns in streamfunction, composites analysis shows the inter-basin interaction from a different perspective than the Walker cells**" near in line 301 of the original manuscript.

- We increase the focus on AP-P (i.e., the influence from the Atlantic forcing on modulating the Pacific ENSO atmospheric teleconnection) and away from the linearity by changing the significance test for figure 4f and 5f from the difference between (AP-P) vs A and towards simply (AP) vs (P) and put all (AP-P) vs A figures in the supplementary section. This also further increases our statistical significance and makes the subfigures more consistent. We also made this change for figures 6, 7, 9, 10 to be consistent.

Fig. 8: You are making comparisons between the correlation coefficients r. I have doubt that some of the differences you describe are statistically significant. I think you should at least check with Fisher z-transformation, find the confidence interval, then transform it back to r. Then you can present each correlation coefficient in a range, based on the significance level you decide. If the uncertainty range overlap between two experiments then the difference is not statistically significant. Here is an example that presents such information in their results:

Revisiting the relationship between jet position, forced response, and annular mode variability in the southern midlatitudes

https://agupubs.onlinelibrary.wiley.com/doi/full/10.1002/2016GL067989

(see their Figure 2).

Thank you for the great suggestion. We indeed did not compare the significant difference between R values, only the significance of the R independently. We have implemented the analysis and results show that the uncertainty ranges are very large, and the intervals overlap between the different correlation values. However, this outcome improves the consistency between our analysis that used changes in R to determine sensitivity and the analysis that uses composite analysis. Therefore, the overall conclusions are not changed in a significant way. To incorporate this new finding into the manuscript by including the following:

- Slight modifications throughout for better reading (see bolding)

- Inclusion of method in the methodology, including: "**We also use correlation (R) throughout, and derive significance by using a confidence interval created by using a Fisher transform (Devore, 1991; Simpson and Polvani, 2016).**"
- Changing the phrase (around line 336 in original manuscript) "When forcing both basins together (AP), the correlation increases by 57% to -0.44 as compared to only forcing the Pacific basin (P)" to now read as: "When forcing both basins together (AP), the correlation increases by 57% to -0.44 as compared to only forcing the Pacific basin (P)**, but we find that the difference is not statistically significant (see confidence intervals in Figure 8a.**"
- We compared the insignificance between the change in the Walker index-RWS with the significant change in RWS anomaly to help further emphasize that the Walker cell may not be the only aspect influencing the relationships
- Point out the insignificance for the Walker-EA relationship, which corresponds well with the proceeding observation that the EA anomaly changes little (lines 356 in original manuscript, 363 in new manuscript)
- The insignificant correlation change also helps make the scatter plots more consistent with the composite analysis, and where possible the writing was updated to reflect this improved consistency.
- In the conclusion we also include: "**Thus, when examining the effectiveness for Atlantic SSTAs to modulate of the extratropical response through using correlation shifts, we do not see a significant strengthening of the link between the Walker cell gradient and either EA or NAO in both boreal spring and summer.**"

Fig. 8: Are the scatters (dots) in each experiment due to internal variability in the same experiment? Or each dot represents one experiment? The SST forcing is the same for a same season year by year, right? I am a little surprised by the 'large' variability of the Walker Index. I thought tropical circulations are much less 'noisy' under the same SST forcing. Would it be useful to add the reanalysis data (under El Nino condition if you like) in these plots for comparison? I understand they won't be then comparing the same things, but it could give us an idea both on what the variability and the relationships between these indices are like in the reanalysis data.

Thank you for sharing this point of concern. Each color represents one experiment type (i.e., A, P, AP). As we previously ran analysis on the basic state, including the seasonal evolution for both JRA-55 and ISCA's climatological experiment, we

believe that the large variability is reasonable and within the bounds of observed internal variability. To help further show this, we included the seasonal evolution of the Walker cell below. As seen in MAM and JJA, +/- 1 stddev has a spread of approximately 5 *10$^{10}$ kg/s for both JRA-55 and ISCA, which is approximately the same as the crosses in figure 8 (horizontal width also represents +/- 1 stddev). Indeed, some parts of the seasonal evolution show that ISCA has a smaller spread ('less noisy'), but this is likely due to the lack of interannual variability.

[Figure]

Lines 440-448: I am not sure this argument works nicey. You are using the observation that there is near-zero RWS from non-zero Walker Index anomaly for P (from Fig. 8a?) and that RWS is related to the Walker Index, to argue that the modulation from the tropical Atlantic in the AP is important to improve relationship of RWS and the Walker Index. But the AP experiment also has a problem of having average near-zero Walker Index anomaly, so how does that result in a non-zero RWS anomaly that is obtained? Maybe there is another factor not considered here and the Walker cell is not suitable factor. Also, you might have mixed up two things

in your descriptions here: the average response of the experiments (compared to Control experiment) and the modulations by the SST forcings on the internally-generated RWS and Walker Index relationships. Or it's possible I have misunderstood Fig. 8, see my question about Fig. 8 above. In any case, I think further improvement in the descriptions and making the arguments is required.

Leading up to line 440, between lines 438 and 440, we explain that we are looking at the correlation, and that there is nearly a doubling of R between P and AP (for the Walker vs. RWS indices). Between lines 440 and 442 explicitly point out that that this strong correlation difference is incongruent with the observation that P has little RWS but a negative Walker cell gradient (line 440, "It is notable that although the Walker index and RWS have a strong relationship, the linear fit does not pass through the origin, whereby the P has a strongly negative Walker index but lacks an anomalous RWS"). This incongruency was also explicitly pointed out the similar issue in the Results section, on lines 346-348 ("Since the mean RWS for P is neutral even with a strongly negative Walker gradient, this indicates that the Walker gradient is not the sole influence on the Caribbean RWS.").

To further improve the description, and to help our argument, we further state that:

**"Finally, to further understand the neutral mean response for P, Supplementary Figure S5 shows the RWS field. We find that only the sensitivity experiments that force the Pacific (P and AP) have a strong negative RWS over the Gulf of Mexico and that this RWS response dominates over the negative RWS over the Caribbean. This RWS may be related to the most southeastern lobe of the PNA pattern, as it strongly overlaps with the Southeastern low presented by Casselman2021 et al., (2021). As a result, even as the Walker cell index may tend towards inducing a positive RWS when forcing the Pacific only, the influence from the Southeastern low may dominate over the influence from the Walker gradient, potentially explaining why the RWS is neutral for P, as opposed to positive."** (around line 350 in the updated manuscript)

Paragraph in Line 464, Fig. 8b, c, and the title of the paper: If I have understood Fig. 8 correctly, I think Fig. 8 is mostly presenting something fundamentally different from Figs. 6, 7 (and 9, 10). For eg, Fig. 8b, c is about how both ENSO and the related tropical North Atlantic SST modulate the relationships between the Walker Index and EA (and also NAO), whereas Figs. 6, 7 instead are mainly about the point of how

ENSO teleconnection is modulated by the tropical North Atlantic (like the paper title). In the former, ENSO is one of the modulators; in the latter, ENSO teleconnection is being modulated. Therefore, Fig. 8 is also not directly related to the title of the paper. (See also the previous comment).

Thank you for your comment. Figures 8b-c are about of the tropical connection to the extratropics following an ENSO event and how it is modulated (moving from the red to the yellow cross) by the presence of the Atlantic SSTAs. To the extent that we look at how the Atlantic SSTAs (independent of Pacific SSTAs) are related to this connection, we primarily use this to help explain why we see the shift from red to yellow.

In this figure, we analyze the changes in the correlation between EA/NAO and the Walker cell with respect to the different SST forcings, showing the modulating role of to the Atlantic SSTAs on the ENSO extratropical connection. Figures 6,7,9,10 use shifts in composite anomalies to show how sensitive the ENSO influence on the extratropical (and tropical for 6,7) is to the Atlantic SSTAs, and to the extent that we look at how the Atlantic SSTAs (independent of Pacific SSTAs) are related to the extratropics, it is to simply help explain why we see the shifts between P and AP. In 6,7 we are actually framing our analysis more towards ENSO perturbing the Walker circulation (including downstream towards the Atlantic) as a teleconnection, from which the Atlantic SSTAs modulate. In Figures 9 and10 we are also framing our analysis more towards ENSO extratropical teleconnection, from which the Atlantic SSTAs modulate. Therefore, we believe that figures 6-10 show and quantify different aspects of the TNA modulation of ENSO teleconnections.

To incorporate your feedback, we have modified the title (which can be found in the minor comments section), and also further justified our use of correlation at the beginning of subsection 3.4, stating: "**We use changes in both the correlation and mean anomaly between *P* and *AP* to quantify how sensitive ENSO's teleconnections are to Atlantic SSTAs**."

**MINOR COMMENTS:**

Title of paper: Most papers on similar subject would be on the cold seasons, if this is about boreal spring and summer, I think it should be reflected in the title.

Thank you for the suggestion, and we agree with the point made. We have changed the title from:

"**Modulation of the ENSO teleconnection to the North Atlantic by the tropical North Atlantic and Caribbean**"

to:

"**Modulation of the El Niño Teleconnection to the North Atlantic by the Tropical North Atlantic during Boreal Spring and Summer**"

Paragraphs in Lines 67-82: Shorten these substantially. They read like an information dump and are not easy to follow. They also appear to disconnect the subsequent and preceding paragraphs, where you are already building up to the aims of the study. So, they might be relocated to an earlier part of Intro.

Thank you for your suggestion. We reduced some of the wording, removed the warm pool section and moved the references to the first paragraph of the results for justifying the JJA season selection. Overall, we were able to shorten it substantially and combine the two paragraphs into one.

Sentences in Lines 88-92: Which seasons/months do you focus on?

Thank you for pointing this out. We have added further information on lines 90 and 91 including: "We consider the inter-actions between the tropical basins, as well as interactions between the tropics and North Atlantic **during boreal spring and summer**."

Sect 2.2 heading: Change to "Model description and experiments".

Thank you for your suggestion, we have changed the title from "Model description" to "Model description and experiments"

Line 122 "We chose to multiply the forcing by 4": Is this a realistic forcing? Would this be an obstacle for using the model results to interpret observed relationships or to understand the real world, considering also there might be the potential issue of nonlinear teleconnections due to forcing amplitudes?

Thanks for your feedback. We are aware of the potential for the nonlinearities, and originally informed the reader on line 122-123 that this forcing was stronger than observations. The rationale for using 4 was to ensure a strong response, but with a clear limitation being that it is more difficult to compare to the real world.

In order to reiterate this, we have added further explanation including: "**Furthermore, the atmospheric response to the SSTA forcing may experience nonlinearities (i.e., upper-level divergence responds nonlinearly to a linear change in the magnitude of Pacific SSTAs), but given our analysis is during boreal spring and summer when Pacific SSTAs are weaker than boreal winter, any nonlinearities may also be weaker than those during boreal winter (Graham and Barnett, 1987; Sabin et al., 2013)** ".

Paragraph in Line 126 and last ever: WHY do you want to "remove the indirect influence from stratospheric variability"? Again, what does this mean in the limitations of using the models results to understand the real world? Isn't it easier to not apply the relaxation in the first place? Perhaps you should also perform experiments without the relaxation now? Why wait for another future study? Are the experiments very expensive to run?

We thank you for your comment and can understand this point of view as it was a major point of discussion on how to format the study. In the end we are aiming for a paper that explicitly focuses on the interactions within the troposphere to better understand the relationship. This choice eliminates the ENSO pathway through the arctic (i.e., the stratospheric pathway), and highlights the pathway through the troposphere, which can occur through the Caribbean. To help emphasize this, we have added:

"**While ENSO teleconnections towards the North Atlantic can travel through either the stratosphere or troposphere (Jiménez-Esteve and Domeisen, 2018), teleconnections that travel through the Caribbean are tropospheric (Hardiman et al., 2019), and hence our study focuses on the tropospheric pathway only.**"

The implications of this choice have already been outlined in the final paragraph of the conclusion (492-498), including how this choice may influence the variability in the North Atlantic, but to a lesser extent as the stratosphere is less active in boreal summer. Furthermore, due to the length of paper that doing both types of studies within the same paper, we have focused on one aspect only.

Line 123 "we aim to determine the importance of the TNA following an El Nino only": Is there a reason choosing to only look at El Nino?

We focus specifically on El Niño to keep the paper within a reasonable length. A complementary study on the interactions during La Niña would also be beneficial and a potential next step beyond this paper.

Line 261 and other places: "Sensitivity experiments": In my view, these are not sensitivity experiments because we don't get a sense of how sensitive the model response is to the change of an input strength.

Thank you for your suggestion. We have replaced the wording "sensitivity experiments" throughout the manuscript following your comment.

Line 334 and other places, "sensitivity runs": Same as above, I would not call these sensitivity runs.

Same reason as stated in the minor comment above.

Figs. 6, 7 and Figs. 9, 10: Is it really essential show both pairs of figures (psi and Z200)? Maybe it's enough to only show results in psi, and reuse the same figures in the description in Sect. 3.4. Extratropical (for a geostrophic argument) geopotential height and psi at 200hPa are related with a factor of the Coriolis parameter.

Thank you for the suggestion, and we have also deliberated over this in the past, as there are indeed many aspects that overlap. However, we feel that although the fields are highly related, the benefit of including both (such as the differences in boreal summer in the NAE region, fig. 7a-b vs. fig. 9a-b) outweighs the drawback of not.

Fig. 8: Have you specifically refer to or describe the dark straight line and the legend for R written in black?

Thank you for pointing this out. This black line was originally the correlation across all simulations, but we have removed the dark line as well as the R written in black as we no longer refer to this in the manuscript.

Fig. 8: Sorry to be pedantic. The correct way to describe these plots is "Y vs. X", not "X vs. Y" as you write.

Thank you for pointing this out, we have changed the subfigures titles in figure 8 to the correct order of "Y vs. X".

Line 382: This description is not precise or not correct. Fig. 9e obviously shows an EA pattern, yet you say it is "less present". Perhaps what you try to say is simply the negative EA pattern in AP is not as strong as the negative EA pattern from A+P (?).

Thank you for pointing out this error, you are indeed correct with your assumption for the meaning, so we have changed the sentence to read as: "The EA pattern in figure 9e indicates that when forcing the two basins simultaneously, the **negative** EA pattern is **weaker** during AP than when linearly adding the two basin forcings (A+P)."

Reviewer #2:

The authors use observational analysis and model experiments to investigate the role of tropical North Atlantic (TNA) in modulating the ENSO teleconnection during boreal spring and summer. The inter-basin relationship between the equatorial Pacific and Atlantic is also examined in this study. This is a potentially constructive contribution to our understanding of how TNA modulates the influence of ENSO on the North Atlantic European region. However, this paper is a little difficult for the reader to follow. There are too many different analyses and indices, but an explanation that ties everything together as a whole story is lacking. I discuss these issues in detail below.

Shorten the introduction. There is too much information and each paragraph is disconnected.

Thank you for the suggestion. We have removed items that we felt were unnecessary, Including:

- Reduced the areas between lines 67-82 and removed warm pool items, and combine two paragraphs in this part of the manuscript
- Around line 26 we also reduced the number of examples for how teleconnections are sensitive as this added
- Lines 46-49 were unnecessary and removed, while the paragraphs surrounding these lines were combined

Line120 "…regression values are multiplied by 4…": The maximum of the SSTA in the Pacific looks too strong (~4°C) and may not appear in the observations. Since you consider the non-linear process of the ENSO influence, too strong ENSO amplitude can lead to an unrealistic ENSO response. The caveats of this approach should be discussed.

Thanks for your feedback. We are aware of the potential for the nonlinearities, and originally informed the reader on line 122-123 that this forcing was stronger than observations. In order to help explain the caveats, we have added further explanation around like 120 including: "**Furthermore, the atmospheric response to the SSTA forcing may experience nonlinearities (i.e., upper-level divergence responds nonlinearly to a linear change in the magnitude of Pacific SSTAs), but given our analysis is during boreal spring and summer when Pacific SSTAs are weaker than boreal winter, any nonlinearities may also be weaker**

**than those during boreal winter (Graham and Barnett, 1987; Sabin et al., 2013)**
".

Section 3.1 and 3.2: I understand that the authors wanted to focus first on the tropical interactions. However, too many features are pointed out here (Walker index, RWS, PCD… ). This makes it difficult for the reader to understand and to know how these features are related to your main question. I suggest reorganizing these two sections and relating these characteristics to your main question before and after the analysis. Why are we discussing the walker index, RWS, PCD…, and what we know from these features?

Thank you for the suggestions. We have made the following changes:

- In section 3.1's final paragraph we further elude towards the PCD and Walker index connection to further explain the connectedness
- In the beginning of 3.2 we further explain why we need to do a correlation, as well as why we look at the PCD, including: "**We compare the PCD to the Walker cell gradient, as it is hypothesized that both mechanisms are highly related.** ".
- At the end of subsection 3.2 we also explain: "**Furthermore, as the Walker gradient and PCD are highly coupled, for the remainder of this paper, we use only the Walker cell gradient.**" Here we are able to reduce the scope and focus on two main tropical interactions—a large scale (Walker gradient) and a local scale (Caribbean RWS).
- The rationale for using the PCD for the first section is to relate to past studies and to validate the hypothesis that the PCD is likely related to the Walker cell gradient.

Lines 208-213: Here is a summary of the previous section. It is better to move this part to the end of the previous section.

Thank you for the great suggestion, we have moved the paragraph to the previous subsection.

Line 216: Although the authors have defined the indices in the Methods, I suggest simply re-stating them in the text when they first appear.

Thank you for your suggestion. For the less common indices (so excluding Nino3.4 and TNA) we included a reminder in brackets for the reader.

Figure 3: Only the upper part of the panel is discussed (TNA, ENSO, PCD lead Walker index). It is unnecessary to show the bottom part of the panel [Walker Index JFM(0)-DJF(0)].

Thank you for pointing this out. We originally discussed the RWS-Walker correlation on lines 234-236, but mistakenly left out the reference to the figure, which we now include.

Lines 238-240: Here is a summary of the previous section. It is better to move this part to the end of the previous section.

Thank you for the feedback. Lines 238-240 relate more towards subsection 3.1 (MLR and Walker cell), so moving it wouldn't be appropriate for the previous subsection (correlation analysis). Therefore, we believe it is best to keep the 2 lines in this area.

Figure 4: Delete the "200 hPa" from the title.

Thank you for pointing this out, we have also removed the 200 hPa from figure 5 for the same error is present.

Line 340 "the Walker response to AP forcing is approximately 0 for both season": This is not correct. It should be "varies around 0".

Thank you for the suggestion, we have now changed the manuscript to use "varies around 0".

Figure 9 & 10: The information in these two figures is almost the same as in Figure 6 & 7. The authors can just use Figure 6 & 7 directly to discuss the ENSO teleconnections.

Thank you for the suggestion, and we have also deliberated over this in the past, as there are indeed many aspects that overlap. However, we feel that although the fields are highly related, the benefit of including both (such as the differences in boreal summer in the NAE region, fig. 7a-b vs. fig. 9a-b) outweighs the drawback of not. For example, streamfunction is much better at showing Matsuno Gill type responses, while struggles to show anomalies over Europe.

Figure 6 & 7: The authors can mark the EA and NAE regions in these figures. This makes it easier for the reader to understand.

Thank you for the suggestion. We have added the EA box in the North Atlantic, but we do not define the NAE region in the paper and therefore also do not put box.

[revised manuscript text omitted]

---

## Referee Report (RR1)

The authors have addressed my comments and the manuscript is suggested to be published after one minor correction.

Please delete the extra 'around' in Line 200.

---

## Author Response (AR2)

Dear Dr. William Roberts,

We express our gratitude to the editor and reviewers for their time and work in evaluating our paper a second time. We identify the significant modifications we made to the paper throughout the editing process. We have addressed the following points in particular:

- The abstract is expanded on to clarify specific concerns
- The introduction is shortened, and we clarified many aspects
- The Results section has had many aspects removed which weren't necessary for the points we were making, and we clarified the motivations for looking into specific topics to help give the paper a better narrative
- For the D&C we have cut, combined, and rearranged many aspects in order to have a more consistent and linear logic and to emphasize our findings more
- Additional non-public comments by the editor have been addressed in the new version of the manuscript

Please find the detailed responses to the reviewer's comments and suggestions below.

Jake Casselman, on behalf of the authors

**Anonymous Referee #1**

While I don't have any more scientific criticism, I find that the writing is still not terribly good. I also don't find it easy to grasp what are the main findings. I don't have time to describe all the problematic parts and to give specific suggestions. But I will give some specific comments focusing on "Abstract" and "Discussion and Conclusion" below.

Please bear in mind that writing in the main body may also need improving although I don't refer to it here.

Thanks for your many suggestions, we decided to go through all sections of the paper. We also put more emphasis on the significance of any description to avoid the feeling that we are simply listing aspects off.

I think the authors need to pay attention to good structures, central themes, flows, and story for all levels (the paper overall, sections, paragraphs, and sentences) in a scientific paper. So, while I could recommend "Acceptance", I think readers will not find the paper nice to read. For this reason, I am specifying "Major revision" to give the authors another chance to improve.

-Section 3
Cut down on anything that was not needed to make the read easier.

Thank you for your suggestion, we have removed anything that wasn't required to prove our conclusions.

1. Abstract and D&C - It's not clearly stated what the typical or expected ENSO teleconnection on the North Atlantic in Spring and Summer is. Then, it's not clearly described what is the modulation from TNA SSTA on the teleconnection is. What property of the TNA SSTA affects what property of the teleconnection? For example, is it warmer TNA SSTA produces a weaker ENSO teleconnection, or even changing the pattern of teleconnection?

Thank you for pointing this out. We have incorporated what would be expected from the ENSO influence in boreal summer into our introduction, explaining how it remains rather unclear. For the abstract, we also emphasized more on what the expected response may be.

2. D&C - It's not very organised at different levels. There are descriptions which follow another that may seem confusing, distracting or even contradictory.

a. First paragraph mentions modulation of the ENSO teleconnection by TNA SST, but doesn't describe what the modulation is (what is the modulation doing to the teleconnection in terms of patterns and strengths).

Thank you for pointing this out, we have added in further explanation which also helps to reconnect the conclusion with out abstract.

Walker cell, Gill response and PCD are then quickly mentioned, but it's not explained whether we should regard them a same interlinked process as far as modulation is concerned or not. In fact, the last sentence says "The PCD is salient for its ability to influence the NAE region via a Rossby wave train, and a potential pathway for the Atlantic to influence the NAE region," which makes me think this is not related to modulation of ENSO teleconnection, but a direct PCD teleconnection to the NAE region. Information/writing like this throughout the paper is confusing.

Thank you for your suggestion, we have rephrased the paragraph to better explain why we looked into the PCD, and also explained the interlinked processes in more detail.

b. Second paragraph starts by telling us Walker cell and PCD are highly correlated and increased understanding can be instrumental for improving seasonal prediction. But then PCD seems to be ignored in the rest of paragraph, and descriptions of Pacific and Atlantic interactions follow. Linear and nonlinear interactions are mentioned, but not described or explained.

Thank you for pointing this out, we have moved up the statement about the PCD and have rearranged the first and second paragraphs. We also removed any unnecessary aspects and were more succinct with our statements.

c. Third paragraph is suddenly a general description of models limitation on simulating tropical atmospheric interactions. The discussion may be useful somewhere in this section or elsewhere. But when I tried to follow summary of main findings in the first two paragraphs, this one suddenly appears like an unwanted distraction.

Thank you for pointing this out. We have moved it to the third last paragraph instead and changed the first half of the paragraph to improve the logical flow from paragraph 2 to 3, and to ensure its clear why they need to know this.

d. Fourth paragraph then mentions yet another factor Caribbean RWS. So now there are TNA SST, Walker cell, Gill response, PCD, and Carribean RWS to keep tract of! Are they all important in your research and in this paper? Can you find a unifying factor between them? Or can you not only selectively report to us the most important thing you find? (and report the others in Appendix/Supplementary). The rest of the paragraph is also not clear or contradictory to the main finding. The Abstract and earlier in this section mention modulation of TNA on ENSO teleconnection, but now this paragraph concludes with "When examining the impact for Atlantic SSTAs to modulate of the extratropical response through using correlation shifts, we do not see a significant strengthening of the link between the Walker cell gradient and either EA or NAO in both boreal spring and summer." So is there a modulation by the Atlantic on the ENSO teleconnection or not!? If you are talking about two different things, you need to be much more careful and clearer in your writing. Like I already commented in the previous review, you need to be clear on what are teleconnections you focus on, what are modulating these teleconnections, what are the modulators, as well as what are the modulations. Then you can also describe the mechanisms for selected important modulations.

Thank you for bringing this to our attention. As we improved the first paragraph to include more description of the modulation, we were able to reduce this paragraph dramatically. This also helped in reducing the contradictory statements, as this paragraph (now third in updated manuscript) focuses more on the EA and NAO, while the relationship with the Walker cell and PCD and Caribbean RWS were in the first paragraph.

e. Fifth paragraph - First sentence talks about composite analysis, second sentence talks about tropical Atlantic to extratropics in spring, third sentence talks about Caribbean to Europe in summer, fourth sentence talks about model experiments - please don't jump around like that, your reviewer (readers will be too) is suffering.

The rest talks about role of Atlantic spring using the model experiments. But the last sentence is "This change between A and AP may help to explain why the Walker cell connection to the EA is not significantly in influenced as the (AP-P) response in the North Atlantic does not overlap with the EA or NAO areas." Importantly, if the response is not in the EA and NAO areas, then why do you conclude there is a modulation by the Atlantic? Maybe you need to explain (again) what is the significance (meaning) of looking at AP-P.

Thank you for pointing these aspects out. As we reduced the length of the third paragraph (previously the fourth), we also combined many of the elements in the fifth into this new paragraph to help with the jumping around. To also help with the jumping nature, we preface the sentence for how it continues from the last.

f. Sixth paragraph - First sentence starts with saying extratropical response to the Atlantic forcing is REDUCED when forced together with the Pacific, and ends with the saying the tropical response is INTENSIFIED when forced alongside the Pacific. The next sentence then says it is unclear what causes the "nonlinearity". Firstly what/which "nonlinearity"? Are you calling the difference in extratropical and tropical responses "nonlinearity"? Secondly and importantly, why are we interested in tropical and extratropical Atlantic responses to Atlantic forcing? Don't you want to focus on responses to Pacific forcing, with and without Atlantic modulation? That's your study focus, it's even in your title! So, again, I don't think you have been clear in your writing (maybe even in your mind) on which teleconnection you focus on and what is/are modulating (meaning modifying) this teleconnection.

Thank you for your suggestions. Indeed, the tropical response intensified while the extratropical response reduced, which was the exact point we were trying to make. We were more explicit with this contradiction, as it was the barrier that caused us to not find the source. By nonlinear, throughout the paper we define it as if AP and A+P are different, as the responses do not linearly add. We have added this in brackets to remind the reader. As for your second point, this paragraph was always talking about the response to adding the Atlantic SSTAs to an ENSO event, and therefore we have tried to clarify this further.

For all limitations, we moved them to the end of the discussion for a more consistent discussion instead of having them scattered throughout.

**Anonymous Referee #2**

The authors have addressed my comments and the manuscript is suggested to be published after one minor correction.

Please delete the extra 'around' in Line 200.

Thanks for pointing this out, we have removed the extra 'around'.